# Towards Adaptive Time Series Foundation Models Against Distribution Shift

## Abstract

Foundation models have demonstrated remarkable success across diverse machine-learning domains through large-scale pretraining. However, their application to time series data poses challenges due to substantial mismatches in the distributions of pretraining datasets. In this paper, we tackle this issue by proposing a domain-aware adaptive normalization strategy within the Transformer architecture. Specifically, we replace the traditional LayerNorm with a prototype-guided dynamic normalization mechanism, where learned prototypes represent distinct data distributions, and sample-to-prototype similarity determines the appropriate normalization layer. This approach effectively captures the diverse characteristics of time series data, ensuring better alignment between pretrained representations and downstream tasks. Our method significantly improves fine-tuning performance, outperforming vanilla pretraining techniques and reducing the negative impact of distribution shifts. Extensive experiments on various real-world time series datasets demonstrate the efficacy of our approach, paving the way for more robust and generalizable time series foundation models.

## 1 Introduction

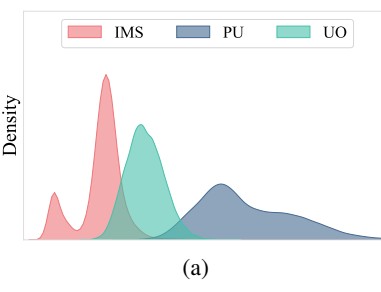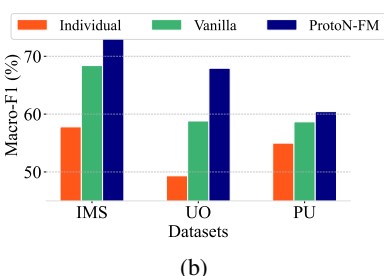

(a)                   (b)

Figure 1: (a) Distributional shifts exist among different time series datasets. (b) Fine-tuning performance comparison on multiple datasets after different pretraining strategies. *Individual* refers to pretraining and fine-tuning a Transformer model on each dataset separately. *Vanilla* denotes pretraining the foundation model on multiple datasets without additional design considerations. In `ProtoN-FM`, we utilize the same multi-dataset pretraining, but incorporate our proposed Domain-Aware Mixture of LayerNorms, resulting in superior performance across diverse datasets.

Foundation Models (FM) have revolutionized machine learning by enabling the learning of general-purpose representations from vast amounts of unlabeled data (Zhou et al., 2023a). These models have achieved remarkable success, particularly in natural language processing (NLP) tasks (Kenton & Toutanova, 2019). In NLP, FMs such as GPT-3 (Brown, 2020), GPT-4 (Achiam et al., 2023), and LLAMA (Touvron et al., 2023) have demonstrated strong performance and generalization capabilities, benefiting from the inherent similarities and structures present in text data.

The ability of FMs to generalize across diverse domains offers promising potential for extending their success into time series (TS) analysis, to be applied to ubiquitous domains such as finance (Yu et al., 2023), healthcare (Moor et al., 2023), and climate (Wu et al., 2023). However, unlike NLP

tasks, where the data distributions are relatively consistent and the models can capture the underlying patterns and semantics, a significant challenge arises when applying FMs to TS data encountered in the mismatch between the data distributions during the pretraining stage (Kim et al., 2021).

This mismatch can be attributed to several factors. First, different TS data often exhibit distinct properties, such as temporal dependencies, irregularities, and domain-specific dynamics. Second, TS data may have varying sampling rates, number of channels, and noise levels, which differ from the clean and well-structured data used in pretraining language models (Wang et al., 2024; Liang et al., 2024). To illustrate this mismatch, Figure 1(a) presents the distribution of various TS datasets for machine fault diagnosis, namely IMS, PU, and UO. The datasets are collected from different engines, and hence, they exhibit significant differences in their value ranges and shapes, highlighting the heterogeneity present in TS data.

The impact of this mismatch on pretraining FMs can be seen in Figure 1(b), which compares the fine-tuning performance upon pretraining with different strategies. We notice that the *Vanilla* pre-training strategy on multiple datasets without considering their heterogeneity achieves suboptimal fine-tuning results. In contrast, considering this mismatch during pretraining achieves better performance, demonstrating the necessity of aligning FMs with the characteristics of TS data.

Therefore, in this work, we propose a novel approach to address the discrepancy between FM pre-training and TS data distributions. Specifically, we introduce a **F**oundation **M**odel design based on **Proto**type-guided dynamic **N**ormalization mechanism (`ProtoN-FM`) within the Transformer archi-tecture, enabling adaptive normalization based on the similarity of samples to learned prototypes, as shown in Figure 2. Unlike traditional LayerNorm, which applies fixed normalization parameters across all samples, our method learns prototypes that capture distinct data characteristics, with each prototype associated with a corresponding LayerNorm module. During training, the model measures the similarity between samples and prototypes, dynamically selecting the most suitable LayerNorm for each sample. This adaptive mechanism allows the model to better align with the heterogeneous nature of TS data, mitigating the distribution shift between pretraining and downstream tasks.

In summary, the main contributions of this work are as follows:

- This is the first work to identify the challenge of data distribution mismatch between foun-dation model pretraining and time series data, which hinders the effective application of foundation models to time series tasks.

- We propose a novel approach introducing a prototype-guided dynamic normalization mech-anism (*ProtoNorm*) within the Transformer architecture, enabling adaptive normalization based on sample similarity to learned prototypes, and capturing domain-specific patterns aligned with time series data characteristics. This layer can be simply included within any Transformer architecture to address the distribution shift during FM pretraining.

- Extensive experiments on diverse real-world time series datasets for different application tasks demonstrate significant fine-tuning improvements and enhanced generalization com-pared to traditional pretraining, highlighting the effectiveness of our proposed approach in identifying the data distribution mismatch.

## 2 RELATED WORK

### 2.1 FOUNDATION MODELS FOR TIME SERIES

Foundation models (FMs) have gained attention in TS analysis, following the success of Large Language Models (LLMs) in natural language processing (NLP) (Liang et al., 2024). However, while some studies have adapted pretrained LLMs for TS data (Cao et al., 2023; Rasul et al., 2024; Gao et al., 2024; Zhou et al., 2023b), this approach is not ideal for TS tasks. The inherent differences between text, which is discrete and categorical, and TS data, which is continuous and numeric, present significant challenges for LLM-based methods (Li et al., 2024). These models often fail to capture the unique temporal patterns and dynamics of TS data. Other research has focused on designing FMs specifically for TS tasks (Das et al., 2024; Liu et al., 2024a; Dong et al., 2024a), often using self-supervised learning techniques like masked sequence prediction (Goswami et al., 2024; Li et al., 2023), contrastive learning (Eldele et al., 2023; Yeh et al., 2023), or hybrid methods (Lee et al., 2024; Dong et al., 2024b). However, it's vital to distinguish works based on their pretraining

strategy. Some methods train on a single dataset and test on that same dataset, such as PatchTST (Nie et al., 2023) and TSLANet (Eldele et al., 2024). While these approaches can achieve strong performance within a specific domain, they do not involve pretraining on multiple datasets, limiting their ability to generalize across diverse TS domains. On the other hand, certain methods adopt a more generalizable approach by pretraining on a pool of datasets (Li et al., 2024; Woo et al., 2024; Ansari et al., 2024), aiming to build foundation models that can generalize well. However, even among these models, some fail to fully address the challenges posed by distribution shifts during pretraining, which can impact their efficacy in real-world applications across different domains.

## 2.2 DISTRIBUTION SHIFTS IN TIME SERIES

Time series data is particularly prone to distribution shifts due to factors such as changes in sensor behavior, environmental variations, and temporal dynamics (Akay & Atak, 2007). A growing body of research aims to mitigate these shifts in deep learning models through techniques such as domain adaptation (Ragab et al., 2023; He et al., 2023; Gong et al., 2024; Ott et al., 2022) and domain generalization (Deng et al., 2024; Lu et al., 2024). These approaches seek to capture domain-invariant features that can be generalized across different distributions. Besides, architecture-specific mechanisms have been developed, including Adaptive RNNs (Du et al., 2021), Non-stationary Transformers (Liu et al., 2022), Instance Normalization flows (Fan et al., 2023; 2024), and contextualized adapters (Chen et al., 2024). These mechanisms aim to alleviate the impact of non-stationary factors through distribution characterization. However, a significant drawback of these designs is their limited transferability across different model architectures, potentially hindering their broader applicability in diverse TS analysis scenarios. Beyond architecture-specific designs, several normalization-based strategies have been proposed to address distribution shifts in TS data (Ogasawara et al., 2010; Passalis et al., 2019). For instance, RevIN (Kim et al., 2021) introduced instance normalization to mitigate distribution shifts by leveraging statistics from individual samples to normalize TS data. Despite these advances, the application of such techniques to Transformer architectures remains limited, and their utilization in multi-dataset training scenarios is still underexplored.

## 2.3 ADAPTIVE NORMALIZATION TECHNIQUES

Adaptive normalization methods, in contrast to traditional fixed schemes, learn flexible strategies to address covariate shift (Vivek Panday, 2022; Fan et al., 2021). For instance, Adaptive Batch Normalization dynamically adjusts normalization parameters across batches, while Adaptive Instance Normalization aligns channel-wise mean and variance to match style input (Li et al., 2018; Chang et al., 2019; Lubana et al., 2021). Recent research has focused on developing adaptive normalization techniques specifically for the non-stationary characteristics of TS data (Deng et al., 2021; Ogasawara et al., 2010). For example, DAIN introduced a non-linear network for adaptive input normalization (Passalis et al., 2019), which was subsequently extended by various approaches (Tran et al., 2021; September et al., 2024). These extensions incorporated adaptive preprocessing layers into deep neural networks. RevIN proposed a symmetric, model-agnostic method that normalizes input sequences and denormalizes model output sequences in TS forecasting (Kim et al., 2021). More recently, SAN introduced slice-level adaptive normalization, offering more flexible normalization and denormalization for TS forecasting (Liu et al., 2024b), while SIN proposed selective and interpretable normalization to select statistics and learn the normalization transformation (Han et al.). While existing normalization methods have shown efficacy, they assume uniform statistical properties across all TS instances, which may not be optimal while pretraining with multiple datasets. In contrast, we explicitly take the distribution inconsistencies into consideration during FM pretraining, offering a more nuanced and effective training strategy.

## 3 PROPOSED METHOD

### 3.1 PRELIMINARIES

#### 3.1.1 PROBLEM DEFINITION

This study addresses the following problem: given a collection of time series datasets $\mathcal{D} = \{\mathcal{D}_k | k = 1, 2, ..., n\}$, where each dataset $\mathcal{D}_k$ contains a variable number of samples with dimensions $L_k \times C_k$

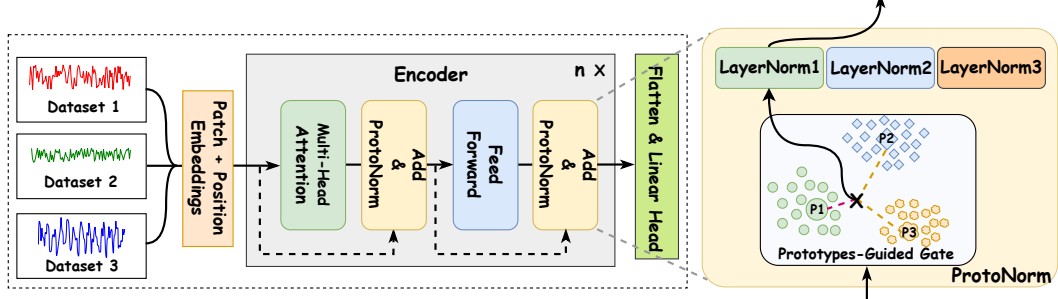

Figure 2: Framework of our proposed `ProtoN-FM`. Input from diverse TS datasets is first partitioned into patches, with positional embeddings added. The resulting output embeddings are then processed through the encoder. Within the encoder, data undergoes normalization using *ProtoNorm* layers, with each comprising two key components: (1) a prototype-guided gate network that matches each sample to the most suitable LayerNorm, and (2) a process that applies the matched LayerNorm for sample normalization.

($L_k$ denoting signal length and $C_k$ representing the number of sensors or variables), our objective is to pretrain a time series foundation model $\mathcal{M}$ on this collection of datasets $\mathcal{D}$ while accounting for inter-dataset distributional shifts. The model is then fine-tuned on either a novel or known dataset using a limited amount of data samples to achieve superior performance.

### 3.1.2 LAYER NORMALIZATION

Layer Normalization (LN) (Ba et al., 2016) is a widely used training technique in deep learning networks, especially in the currently prevalent Transformer architecture (Vaswani, 2017). Instead of normalizing across the batch dimension, i.e., Batch Normalization (BN), LN normalizes across the features within a single layer. Similar to BN, LN also has two trainable affine parameters $\gamma$ and $\beta$ to allow the network to learn different scales and shifts. Given a layer's activation $x \in \mathbb{R}^{C \times L}$ for a single input, LN is expressed as follows,

$$LN\left(x_i; \gamma, \beta\right) = \gamma \cdot \hat{x}_i + \beta, \tag{1}$$

where

$$\hat{x}_i = \frac{x_i - \mu}{\sqrt{\sigma^2 + \epsilon}}. \tag{2}$$

The $\mu$ is the mean and $\sigma^2$ is the variance computed over the features of the layer for a single input, which are denoted as,

$$\mu = \frac{1}{d}\sum_{i=1}^{d} x_i, \quad \sigma^2 = \frac{1}{d}\sum_{i=1}^{d}\left(x_i - \mu\right)^2, \tag{3}$$

and $\epsilon$ is a small constant to avoid divide-by-zero.

In the training phase, LN computes the mean and variance across the features of a single training example at each layer. The normalization step helps to stabilize the learning process by reducing the internal covariate shift. During the testing phase, LN behaves almost identically to the training phase. The difference is that the model is no longer learning or updating the parameters, so the role of LN is purely to normalize the activations and apply the learned scaling and shifting.

Since LN normalizes the features of each sample rather than the batch, there is no need to accumulate running statistics as in BN. This makes LN consistent between the training and testing phases, with no discrepancies between the statistics computed during training and those used during inference.

### 3.2 PROTOTYPE-GUIDED DYNAMIC NORMALIZATION MECHANISM

It is important to explain why we chose to modify LN specifically, rather than other components of the Transformer, to address the distribution shift problem. LN is an ideal candidate for this modification because it has fewer parameters than other parts of the Transformer, making it computationally

efficient to replicate. This allows us to handle variations across different datasets while minimizing the risk of overfitting.

In addition, previous research has demonstrated that domain-specific normalization techniques, such as BatchNorm, are highly effective in reducing domain shifts in adaptation tasks (Chang et al., 2019). This success inspired us to explore a domain-aware normalization strategy tailored for time series data. However, traditional LN approaches assume a static relationship between input samples and their corresponding normalization strategies, potentially limiting the model's adaptability to both intra- and inter-dataset variations. Relying on a fixed normalization strategy for an entire dataset may fail to address challenges such as mixed sample characteristics or cross-domain overlap.

**Prototype-Guided Gating Network.** To overcome these limitations, we introduce `ProtoN-FM`, which implements an adaptive and dynamic normalization mechanism, as illustrated in Figure 2. Rather than employing a fixed LN for each dataset, we propose *ProtoNorm* layer, which consists of multiple LN modules, where one of them is selected based on a prototype-guided gating network that matches each sample to the most appropriate LN based on its proximity to learned prototypes. Upon the completion of the pretraining, the learned prototypes act as anchors that represent different data distributions, allowing the model to adapt its normalization strategy on a per-sample basis.

Figure 3 demonstrates how these learned prototypes function as centroids or representative anchors, capturing distinct data distributions. This approach enables the model to flexibly select the optimal normalization strategy for each sample, thereby accommodating subtle variations within and across datasets, and enhancing its capacity to handle complex or overlapping data distributions.

Formally, each *ProtoNorm* layer predefines a set of $n$ LayerNorm modules $\{LN_1, LN_2, ..., LN_n\}$, alongside a prototypes-guided gating network $\mathcal{G}$. Considering a TS signal $v$ and its features $x$, $\mathcal{G}$ determines which LayerNorm contributes to the input's normalization. Specifically, $\mathcal{G}$ computes the distance between $x$ and a set of predefined prototypes $\{p_1, p_2, \ldots, p_n\}$, each corresponding to one LayerNorm.

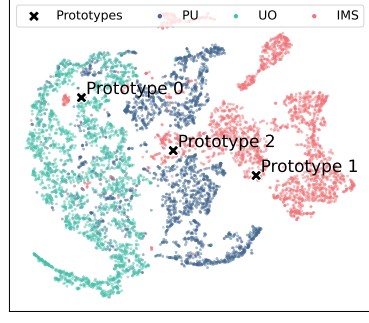

**Adaptive Normalization.** The network selects the Layer-Norm module $LN_i$ whose prototype $p_i$ minimizes the distance to $x$, matching the input to the most suitable normalization function. This selection is given by:

$$i^* = \arg \min_{i \in \{1,2,...,n\}} d(x, p_i), \qquad (4)$$

where $d(x, p_i)$ represents the distance metric (e.g., Euclidean distance) between $x$ and prototype $p_i$.

Figure 3: Visualization of learned prototypes and sample features. Prototypes capture the unique distribution patterns of each cluster.

**Prototype Update.** The prototypes are updated during training using Exponential Moving Average (EMA) (Kingma, 2014), ensuring gradual adaptation based on the evolving input distributions. Formally, the prototype $p_i$ is updated as:

$$p_i^{(t+1)} = \alpha \cdot p_i^{(t)} + (1 - \alpha) \cdot x, \qquad (5)$$

where $p_i^{(t)}$ is the prototype at time $t$, $x$ is the current input feature, and $\alpha$ is the EMA decay factor. This update process ensures that prototypes evolve to better represent the underlying data distributions throughout training, maintaining robustness and adaptability.

**Orthogonality Constraint.** To ensure the learned prototypes remain distinguishable, we introduce an additional orthogonality constraint. Initially, the prototypes are initialized with orthogonality parameters, enabling the gating network to better differentiate among diverse input features and distributions. Further, we implement a regularization technique that encourages prototype independence by minimizing their deviation from orthogonality, inspired by (Saito et al., 2017). Formally, given a matrix $P \in \mathbb{R}^{n \times d}$ where each row represents a prototype, we define the orthogonal loss as:

$$\mathcal{L}_{\text{orth}} = \|PP^T - I\|_F^2 \qquad (6)$$

where $I$ is the identity matrix, and $\| \cdot \|_F^2$ denotes the Frobenius norm.

## 3.3 SELF-SUPERVISED PRETRAINING

We pretrain the foundation model using an augmentation-based contrastive learning approach for time series modeling. This procedure uses augmented versions of time series data to learn robust feature representations. Given an input TS sample $x$, we apply two augmentation techniques: time-shift and scaling with jitter (Eldele et al., 2023), generating two diverse views of the same sample, denoted as $\widetilde{x}_1$ and $\widetilde{x}_2$. Time-shift augmentation introduces variations in signal timing by shifting the input sequence along the temporal axis, while scaling with jitter applies random scaling factors combined with small perturbations, simulating variability in signal amplitude and sensor noise.

Both $\widetilde{x}_1$ and $\widetilde{x}_2$ are processed by the encoder and projector head to produce representation vectors $z_1$ and $z_2$, respectively. We then employ the NT-Xent loss (Chen et al., 2020) to maximize similarity between different views of the same sample while minimizing similarity with other samples. For a batch of $N$ samples, the NT-Xent loss for each augmented pair $(\widetilde{x}_1, \widetilde{x}_2)$ is defined as:

$$\mathcal{L}_{\text{NT-Xent}} = -\log \frac{\exp(\text{sim}(z_1, z_2)/\tau)}{\sum_{j=1}^{2N} \mathbf{1}_{[j \neq i]}\exp(\text{sim}(z_i, z_j)/\tau)}, \tag{7}$$

where $\text{sim}(z_1, z_2) = \frac{z_1 \cdot z_2}{\|z_1\|\|z_2\|}$ denotes the dot product between $\ell_2$ normalized $z_1$ and $z_2$ (i.e., cosine similarity), $\tau$ is a temperature scaling parameter, and $\mathbf{1}_{[j \neq i]}$ is an indicator function excluding the positive pair from the denominator.

The complete loss function for pretraining incorporates both the contrastive learning loss and the orthogonal loss, ensuring robust representation learning and distinct, separable prototypes. We express the total loss as:

$$\mathcal{L} = \mathcal{L}_{\text{NT-Xent}} + \lambda \cdot \mathcal{L}_{\text{orth}} \tag{8}$$

where $\lambda$ is a hyperparameter that balances the contribution of the orthogonal loss in the overall optimization. In this study, we empirically set $\lambda$ to 0.001.

## 4 EXPERIMENTS

This section evaluates the effectiveness of our proposed method across diverse real-world time series classification tasks. We present the primary results of our approach to fault diagnosis (FD) and human activity recognition (HAR) tasks. Subsequently, we conduct an ablation study and analyze key model parameters. Finally, we provide an extended analysis of the model's performance and behavior, offering a comprehensive assessment of our method's capabilities and limitations. To promote reproducibility and further research, the implementation code will be made publicly available.

### 4.1 EXPERIMENTAL SETUP

**Datasets.** We demonstrate the advantages of the proposed method on two key application tasks: FD and HAR. Specifically, for the FD task, we employ six datasets (i.e., IMS (Qiu et al., 2006), UO (Huang & Baddour, 2018), PU Lessmeier et al. (2016), CWRU (Smith & Randall, 2015), FEMTO (Nectoux et al., 2012), and XJTUSY (Wang et al., 2020)) for pretraining phase. Subsequently, we fine-tune and evaluate the model's performance on three datasets (i.e., IMS, UO, PU). In the HAR task, we utilize five datasets (i.e., HHAR (Stisen et al., 2015), SKODA (Stiefmeier et al., 2008), UCIHAR (Anguita et al., 2013), USCHAD (Zhang & Sawchuk, 2012), and WISDM (Kwapisz et al., 2011)) for pretraining phase, followed by fine-tuning and performance evaluation on each individual dataset. Detailed information regarding data preprocessing procedures and dataset characteristics is provided in Appendix A.

**Handling Varying Time Series Characteristics.** Due to the variability among TS datasets, we implement the following preprocessing. First, we fix the varying numbers of channels by repeating the channels in samples with fewer channels to match the maximum channel count in the whole pretraining dataset pool. Notably, we mitigate potential overfitting to artificially duplicated data by introducing random noise to these repeated channels. Second, we standardize sequence lengths across samples by employing a two-pronged approach: longer sequences are downsampled to the target length, while shorter sequences are zero-padded to reach the desired length. Specifically, we standardize sequence lengths to 1024 for FD tasks and 128 for HAR tasks. These preprocessing techniques ensure uniform input dimensions, enabling our model to train effectively.

**Model Architecture.** We adopt the PatchTST architecture (Nie et al., 2023) for its simplicity and effectiveness. The input data is initially segmented into patches, which are then mapped to embeddings. These embeddings are then processed by the encoder to extract salient features. The encoder comprises multiple layers, each constructed with a multi-head attention mechanism followed by a *ProtoNorm* layer, and a feed-forward network succeeded by another *ProtoNorm* layer, as depicted in Figure 2. During the pretraining phase, the encoder-extracted features are directed to the contrastive learning head for self-supervised training. In the fine-tuning and testing phases, these features are instead fed into a classification head, consisting of linear classifiers, to generate predictions.

**Hyperparameters.** We optimize our model using the AdamW optimizer with a learning rate of $1e-3$, weight decay of $1e-5$, and dropout rate of $0.15$. A cosine learning rate schedule with $2000$ warmup steps is applied across all tasks. For the FD task, we employ a pretraining batch size of $256$ over $5$ epochs, with an embedding dimension of $256$, $8$ attention heads, $12$ encoder layers, a patch size of $50$, and an input sequence length of $1024$; fine-tuning maintains this architecture but reduces the batch size to $64$ and extends training to $50$ epochs. HAR task uses a pretraining batch size of $128$ over $5$ epochs, with an embedding dimension of $128$, $8$ attention heads, $6$ encoder layers, a patch size of $32$, and an input sequence length of $128$; fine-tuning reduces the batch size to $8$ and extends training to $50$ epochs. Model performance was evaluated using accuracy and macro-averaged F1 scores as primary metrics. Each experiment was repeated three times, with the average performance reported. The method was implemented using PyTorch and conducted on NVIDIA L40 GPUs.

**Baselines and Training Protocol.** We benchmark our method against supervised training (*Sup.*), pretraining on individual datasets (*Individual*), and conventional pretraining across multiple datasets (*Vanilla*). For each application task, we fine-tune the model on ***100 randomly selected samples per dataset***. However, for datasets with a high number of classes, we ensure a minimum of 5 samples per class, even if this exceeds 100 total samples for that dataset. We initialize the model with pretrained weights and replace the self-supervised learning head with a linear classifier. The model is then fine-tuned on the downstream dataset, optimizing the learned representations for effective generalization with minimal labeled data. During this fine-tuning stage, all prototypes remain frozen. Finally, we evaluate the model's performance using the test set from each respective dataset.

## 4.2 EXPERIMENTAL RESULTS

**Performance Comparison on FD Task.** Table 1 demonstrates the efficacy of the proposed `ProtoN-FM` method compared to three baseline approaches across FD tasks. `ProtoN-FM` outperforms all other methods, achieving an average accuracy of 70.33% and an average Macro-F1 score of 67.13%. Notably, all self-supervised learning pretraining methods surpass supervised training, underscoring their ability to capture complex patterns and variations inherent in time series data, thus enabling richer feature representations. Furthermore, pretraining on multiple datasets exhibits enhanced performance compared to individual dataset pretraining, suggesting that the incorporation of diverse data facilitates more robust representation learning. While the Vanilla method shows improvement over individual pretraining, it fails to account for distribution shifts between datasets, limiting its performance relative to `ProtoN-FM`. This finding validates that by explicitly addressing these shifts through a prototype-guided dynamic normalization mechanism, `ProtoN-FM` effectively aligns its learning process with the heterogeneity present in real-world time series data.

Table 1: Performance comparison of various methods on FD task. We calculate the Accuracy and F1-score (%) for each dataset. The best results are **bolded** and the second best results are underlined.

| Datasets | Accuracy | | | | Macro-F1 | | | |
|---|---|---|---|---|---|---|---|---|
| | Sup. | Individual | Vanilla | `ProtoN-FM` | Sup. | Individual | Vanilla | `ProtoN-FM` |
| IMS | 54.22 | 59.48 | 77.00 | **78.78** | 47.84 | 57.79 | 68.39 | **73.03** |
| UO | 49.32 | 50.62 | 60.00 | **68.56** | 48.20 | 49.33 | 58.81 | **67.93** |
| PU | 48.19 | 58.42 | 61.91 | **63.65** | 44.61 | 54.98 | 58.66 | **60.43** |
| Average | 50.58 | 56.17 | 66.30 | **70.33** | 46.88 | 54.03 | 61.95 | **67.13** |

Table 2: Performance comparison of various methods on HAR task. We calculate the Accuracy and F1-score (%) for each dataset. The best results are **bolded** and the second best results are underlined.

| Datasets | Accuracy | | | | Macro-F1 | | | |
|---|---|---|---|---|---|---|---|---|
| | Sup. | Individual | Vanilla | ProtoN-FM | Sup. | Individual | Vanilla | ProtoN-FM |
| HHAR | 69.57 | 70.23 | 71.07 | **72.43** | 61.44 | 62.67 | 63.08 | **64.36** |
| SKODA | 17.76 | 23.48 | 22.52 | **25.56** | 11.64 | 15.27 | 14.69 | **16.94** |
| UCIHAR | 54.01 | 55.68 | 57.69 | **59.38** | 43.03 | 44.61 | 45.54 | **46.69** |
| USCHAD | 30.52 | 32.01 | 34.69 | **36.64** | 18.73 | 20.45 | 22.14 | **23.86** |
| WISDM | 54.61 | 55.74 | 58.16 | **61.25** | 37.56 | 38.23 | 40.32 | **42.67** |
| Average | 45.29 | 47.43 | 48.83 | **51.05** | 34.48 | 36.25 | 37.15 | **38.90** |

**Performance Comparison on HAR Task.**   Table 2 presents the classification performance analysis for HAR tasks. The proposed ProtoN-FM method demonstrates superior efficacy compared to baseline approaches, achieving an average accuracy of 51.05% and an average Macro-F1 score of 38.90%. Consistent with the findings in FD tasks, all self-supervised learning pretraining methods outperform supervised training. ProtoN-FM consistently surpasses the Vanilla approach, which, despite showing improvements over individual pretraining, fails to adequately address distribution shifts between datasets. This underscores the importance of incorporating diverse training data and accounting for the heterogeneity inherent in real-world HAR tasks. These performance metrics confirm that ProtoN-FM not only enhances classification accuracy but also provides a more nuanced understanding of the underlying data dynamics in HAR applications.

## 5   MODEL ANALYSIS

### 5.1   ABLATION STUDY

Table 3 evaluates the contribution of different model components, comparing the average performance of ProtoN-FM against two variants across various datasets in the FD task. The *w/o ProtoGate* variant represents the domain-specific LayerNorm model (i.e., DSLN), which replaces the prototype-guided gate network with dataset-specific LayerNorm selection. A more detailed description of this variant method is provided in Appendix B. The *w/o OrthoConstrain* variant indicates the orthogonality constraints are omitted from the model. The experimental results demonstrate that removing the prototype-guided gate network (i.e., *w/o ProtoGate*) yields a notable decline in performance, underscoring its crucial role in dynamically matching appropriate data distributions. In contrast, using a fixed LayerNorm for each dataset may overlook subtle intra-dataset variations. Additionally, removing the orthogonality restrictions (i.e., *w/o OrthoConstrain*) also diminishes model performance, suggesting that imposing a separation constraint on learned prototypes enables the gating network to better differentiate among diverse input features and distributions.

Table 3: Ablation study to the effect of each component. We calculate the Accuracy and F1-score (%) for each dataset. The best average performance results are **bolded**.

| Variants | Accuracy | | | | Macro-F1 | | | |
|---|---|---|---|---|---|---|---|---|
| | IMS | UO | PU | Average | IMS | UO | PU | Average |
| w/o ProtoGate | 77.51 | 60.43 | 62.02 | 66.65 | 69.26 | 59.37 | 58.81 | 62.48 |
| w/o OrthoConstrain | 77.53 | 66.99 | 63.80 | 69.44 | 70.51 | 66.22 | 60.69 | 65.81 |
| ProtoN-FM | 78.78 | 68.56 | 63.65 | **70.33** | 73.03 | 67.93 | 60.43 | **67.13** |

### 5.2   PARAMETER ANALYSIS

We conduct parameter analyses of our model, focusing on two key parameters: the number of LayerNorms per *ProtoNorm* layer and the orthogonal loss weight $\lambda$. For the former, we compare models with $\{2, 3, 4, \#D\}$ LayerNorms, where $\#D$ represents the number of pretraining datasets. For $\lambda$, we

evaluate performance across values of $\{0.001, 0.01, 0.1, 1\}$. This systematic exploration allows us to assess the impact of these parameters on model performance and identify optimal configurations.

**Effect of Number of LayerNorms.** Figure 4 illustrates the average performance of `ProtoN-FM` with varying numbers of LayerNorms within each *ProtoNorm* across different datasets for both FD and HAR tasks. Detailed results for each dataset are provided in Appendix C.1. For the FD task, employing three LayerNorms within per *ProtoNorm* layer yields optimal performance, with an average accuracy of 70.33% and a Macro-F1 score of 67.80%, marginally outperforming other configurations. Conversely, in the HAR task, setting the number of LayerNorms equal to the number of pretrained datasets ($\#D$) achieves the highest performance, with an accuracy of 51.85% and a Macro-F1 score of 38.50%. These findings suggest that the optimal number of LayerNorms may be task-dependent, with a slight advantage in matching the LayerNorm count to the number of datasets in more diverse or complex tasks such as HAR. Notably, the model's performance remains relatively stable across different numbers of LayerNorms, indicating robustness to this parameter choice.

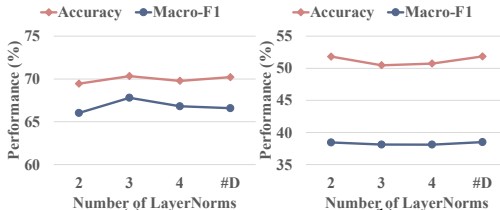
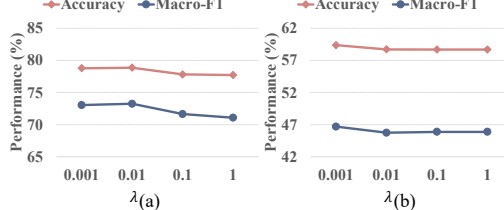

Figure 4: Average performance comparison with varying number of LNs across different datasets on different tasks. (a) Average performance on different datasets of FD task. (b) Average performance on different datasets of HAR task.

Figure 5: Average performance comparison using varying number of $\lambda$ across IMS dataset of FD task and UCIHAR dataset of HAR task. (a) Average performance on IMS of FD task. (b) Average performance on UCIHAR of HAR task.

**Effect of Orthogonal Weights $\lambda$.** Figure 5 illustrates the effect of varying the orthogonal loss weight $\lambda \in \{0.001, 0.01, 0.1, 1\}$ on the IMS (in the FD task) and UCIHAR (in the HAR task) datasets. For the IMS dataset, $\lambda = 0.01$ yields optimal performance, achieving an accuracy of 78.86% and a Macro-F1 score of 73.25%. Performance declines as $\lambda$ increases, suggesting that larger values may lead to over-regularization. Conversely, the UCIHAR dataset exhibits less sensitivity to $\lambda$, with only minor fluctuations in both accuracy and Macro-F1. The highest accuracy of 59.38% and Macro-F1 of 46.69% are observed at $\lambda = 0.001$, but overall performance remains stable across different values. These results indicate that model performance is not highly sensitive to $\lambda$, and smaller values tend to suffice for optimal performance across both tasks.

## 5.3 GENERALIZATION ANALYSIS

This section evaluates the generalization capacity of the proposed `ProtoN-FM` model in comparison to the Vanilla pretraining method across both fault diagnosis and human activity recognition tasks. For each dataset within each task, we employ a cross-domain pretraining approach: the model is pretrained on all datasets except the target dataset, fine-tuned on a small portion of the target dataset, and subsequently tested on its corresponding test set. Figure 6 illustrates the average performance across different datasets for both FD and HAR tasks. Comprehensive results for individual datasets are provided in Appendix C.2.

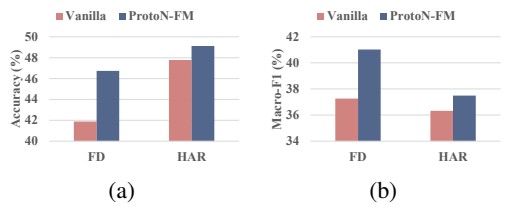

Figure 6: Average generalization performance of `ProtoN-FM` and Vanilla on FD and HAR Tasks. (a) Comparison of the average accuracy on all datasets of each task. (b) Comparison of the average macro-F1 on all datasets of each task.

Figure 6 demonstrates that `ProtoN-FM` consistently outperforms Vanilla pretraining in both accuracy and Macro-F1 scores across both appli-

cation tasks. In the FD task, `ProtoN-FM` achieves a notable improvement, increasing accuracy from 41.89% to 46.73% and Macro-F1 from 37.27% to 41.03%. Similarly, for the HAR task, `ProtoN-FM` surpasses Vanilla pretraining with an accuracy gain from 47.78% to 49.12% and a Macro-F1 boost from 36.32% to 37.49%. These results underscore the enhanced generalization ability of `ProtoN-FM`, particularly in handling distribution shifts across diverse datasets. The consistent improvements across both application domains highlight the model's robustness and efficacy in capturing meaningful representations during pretraining, which are better aligned for fine-tuning on new, unseen datasets.

## 5.4 ANALYSIS OF VARYING DISTRIBUTION SHIFTS

In this section, we evaluate the effectiveness of our `ProtoN-FM` model under varying levels of distribution shifts using the IMS dataset. To simulate different shift magnitudes, we create three perturbed versions of the IMS dataset (i.e., IMS-N1, IMS-N2, IMS-N3) by adding Gaussian noise with increasing standard deviations (i.e., 0.1, 0.2, and 0.3, respectively). The model is pretrained on paired datasets (i.e., IMS with IMS-N1, IMS-N2, or IMS-N3) and fine-tuned with a small subset of IMS data. Performance is assessed on the IMS test set, comparing our `ProtoN-FM` method with the Vanilla pretraining approach. As illustrated in Figure 7, `ProtoN-FM` consistently outperforms Vanilla pretraining across all perturbation levels. Notably, in the most challenging scenario (i.e., IMS-N3), `ProtoN-FM` improves accuracy from 67.05% to 70.54% and Macro-F1 from 63.84% to 66.28%, demonstrating its robustness in handling distribution shifts across the data.

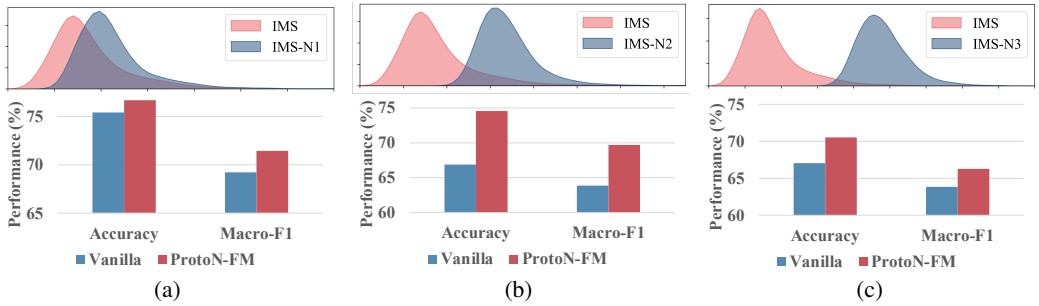

Figure 7: Comparative average performance of `ProtoN-FM` and Vanilla method on the IMS Dataset under varying distribution shifts. IMS-N1, IMS-N2, and IMS-N3 represent increasingly perturbed versions of the original IMS dataset. (a) Pretraining on IMS and IMS-N1, fine-tuning and testing on IMS. (b) Pretraining on IMS and IMS-N2, fine-tuning and testing on IMS. (c) Pretraining on IMS and IMS-N3, fine-tuning and testing on IMS.

## 6 CONCLUSION

This paper introduces `ProtoN-FM`, a novel approach addressing the discrepancy between foundation model pretraining and time series data distributions. `ProtoN-FM` enables adaptive normalization based on the similarity of samples to learned prototypes. Unlike traditional LayerNorm, which applies fixed normalization parameters across all samples, our method learns prototypes that capture distinct data characteristics, with each prototype associated with a corresponding LayerNorm module. Comprehensive experiments across diverse datasets in various application classification tasks demonstrate our superior performance over traditional Transformer design, particularly in alleviating data distribution mismatches in time series data. Future research should explore the universal capabilities of our model in handling additional downstream tasks, such as forecasting and anomaly detection. Moreover, integrating *ProtoNorm* layer into various Transformer architectures to fully leverage its pretraining potential presents a promising avenue for future work.

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

# A  DATASETS DETAILS

## A.1  DATA PREPROCESSING

For all application tasks, we employ a $60/20/20$ ratio for train/validation/test splits. In addition, to enhance the scale and diversity of our data, we incorporate three additional prognostics and health management (PHM) datasets—CWRU, FEMTO, and XJTUSY—to augment the pretraining phase for FD tasks. Subsequently, we fine-tune and evaluate the model's performance on three datasets (i.e., IMS, UO, PU). In the HAR task, we utilize five datasets (i.e., HHAR, SKODA, UCIHAR, USCHAD, and WISDM) for pretraining phase, followed by fine-tuning and performance evaluation on each individual dataset.

## A.2  FD TASK

Our fine-tune and evaluate on the FD task employs three datasets, described as follows:

- **IMS:** This dataset, sourced from the University of Cincinnati, comprises data from three run-to-failure experiments conducted on a loaded shaft. The experimental setup consisted of a shaft supported by four roller bearings, with each bearing housing instrumented with both vertical and horizontal accelerometers. The experiments culminated in the development of a defect on one of the bearings, providing valuable data on the progression of bearing failure under controlled conditions.

- **UO:** This dataset encompasses vibration signals from bearings operating under diverse health conditions and rotational speeds. It comprises 36 signal sets, each corresponding to one of 12 experimental conditions derived from combinations of three bearing health states (healthy, inner race defect, outer race defect) and four rotational speed patterns (ascending, descending, ascending-descending, and descending-ascending). To ensure data reliability, three trials were conducted for each condition. In the UO dataset, the bearing's health state serves as the class label, while the various rotational speed patterns represent distinct domains.

- **PU:** This dataset comprises vibration signals from an electric motor, encompassing 32 distinct signal sets, each corresponding to a different bearing. The bearings are categorized as follows: 6 healthy, 12 with artificial damage, and 14 with real damage incurred under actual operating conditions. Each bearing was subjected to four different working conditions. In this dataset, the bearing type serves as the class label, while the various operating conditions represent distinct domains. For our study, we focused on the data collected from real damaged bearings across all working conditions to conduct performance verification.

To augment the pretraining data for the FD task, we incorporate three additional related prognostics and health management datasets.

- **CWRU:** This dataset, provided by the Case Western Reserve University Bearing Data Center, comprises vibration signals collected at frequencies of 12 kHz or 48 kHz from both normal bearings and those with single-point defects, under four distinct motor load conditions. For each operational state, single-point faults were artificially induced on the rolling element, inner ring, and outer ring, with fault diameters of 0.007, 0.014, and 0.021 inches, respectively. In our study, we utilized data collected from the drive end, sampled at 12 kHz.

- **FEMTO:** This dataset, sourced from the FEMTO-ST Institute in France, comprises 17 accelerated run-to-failure experiments. Acceleration and temperature data were collected using a test bench that subjected bearings to variable loads and speeds under three distinct operating conditions. Data acquisition occurred at 10-second intervals, with each sample spanning 0.1 seconds. This experimental design provides a comprehensive dataset for studying bearing degradation under controlled, accelerated conditions.

- **XJTUSY:** This dataset comprises run-to-failure data from fifteen bearings, tested under three distinct operating conditions. Each recording consists of 32768 data points, captured using a dual-channel vibration sensor sampling at 2.56 kHz. The data acquisition protocol involved recording 1.28-second snapshots at one-minute intervals, providing a comprehensive time series of bearing degradation across varying operational parameters.

A summary of the characteristics of these two types datasets is presented in Tables 4 and 5.

Table 4: A description of characteristics of the datasets on FD task used in our experiments.

| Dataset | # Train | # Test | Length | # Channel | # Class |
|---------|---------|--------|--------|-----------|---------|
| IMS | 42492 | 14164 | 20480 | 1 | 4 |
| UO | 42184 | 14061 | 2000000 | 2 | 3 |
| PU | 163296 | 54432 | 249600 | 1 | 14 |

Table 5: A description of characteristics of the datasets on other related PHM datasets.

| Dataset | # Train | # Val | Length | # Channel | Sample Rating |
|---------|---------|-------|--------|-----------|---------------|
| CWRU | 280 | 2201 | 120000 | 2 | 12kHz |
| FEMTO | 11794 | 11934 | 2560 | 2 | 25.6kHz |
| XJTUSY | 191040 | 202912 | 32768 | 2 | 25.6kHz |

## A.3 HAR TASK

Our evaluation of the HAR task employs five datasets, described as follows:

- **HHAR:** This dataset is distinguished by its diverse data collection methodology, encompassing multiple device types (smartphones and smartwatches) and various individuals performing a range of activities including cycling, sedentary postures (sitting and standing), ambulation (walking), and stair navigation. The dataset's heterogeneity in terms of device types and body positions presents a challenging benchmark for evaluating model generalization across diverse sensor configurations and activity categories.

- **SKODA:** This dataset is specifically designed to monitor worker activity in a manufacturing assembly-line environment. It comprises accelerometer readings from ten distinct positions on subjects' arms, with each data point annotated with a specific activity class, including a null class. The multi-point sensor placement and task-specific labeling make this dataset particularly valuable for studying fine-grained human activities in industrial settings. For this dataset, we selected 5 out of 113 channels for our experiments, i.e., channels with ID 55, 45, 52, 50, and 58. Those were selected based on the correlations between channels.

- **UCIHAR:** This dataset comprises experimental data collected from a cohort of 30 volunteers performing six activities: walking, ascending stairs, descending stairs, sitting, standing, and lying down. Participants wore a waist-mounted smartphone equipped with embedded accelerometer and gyroscope sensors, which captured 3-axial linear acceleration and 3-axial angular velocity data, respectively.

- **USCHAD:** This dataset comprises six-dimensional readings from body-worn 3-axis accelerometers and gyroscopes, collected via Motion-Node devices. The study population consists of 14 subjects (7 male, 7 female), balanced for gender and with specified physical characteristics and age demographics. Data were sampled at 100 Hz, with each time-step annotated with one of 12 distinct activity class labels. This comprehensive and well-structured dataset provides a robust foundation for human activity recognition research.

- **WISDM:** This dataset comprises time series data collected from smartphone sensors and wearable devices, capturing a diverse range of human activities including ambulation (walking and jogging) and stationary postures (sitting and standing). The heterogeneous nature of the user-generated activity data renders this dataset particularly suitable for evaluating the robustness of HAR models across varied motion patterns and sensor placements.

A summary of the characteristics of these datasets is presented in Table 6.

Table 6: A description of characteristics of the datasets on HAR task used in our experiments.

| Dataset | # Train | # Test | Length | # Channel | # Class |
|---------|---------|--------|--------|-----------|---------|
| HHAR    | 10233   | 4436   | 128    | 3         | 6       |
| SKODA   | 2919    | 974    | 177    | 5         | 10      |
| UCIHAR  | 5881    | 2947   | 128    | 9         | 6       |
| USCHAD  | 2992    | 1008   | 500    | 6         | 12      |
| WISDM   | 6309    | 2104   | 100    | 3         | 6       |

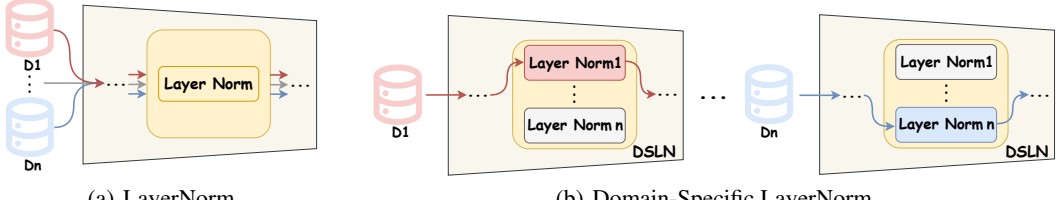

(a) LayerNorm

(b) Domain-Specific LayerNorm

Figure 8: Comparison of LayerNorm (LN) and Domain-Specific LayerNorm (DSLN). A DSLN layer comprises $n$ branches, each corresponding to a specific dataset. Input signals are directed to the appropriate branch based on their attributed dataset.

## B  DOMAIN-SPECIFIC LAYER NORMALIZATION

Domain-Specific LayerNorm (DSLN), a variant of our model, is implemented by using multiple sets of LNs reserved for each time series dataset. Figure 8 illustrates the difference between LN and DSLN. Formally, DSLN allocates domain-specific affine parameters $\gamma^k$ and $\beta^k$ for each dataset $\mathcal{D}_k \in \mathcal{D}$. Let $x^k = \{x_i^k | i = 1, 2, ..., d; k = 1, 2, ..., n\}$ denotes a layer's activation for a single input belong to dataset $\mathcal{D}_k$, then DSLN layer can be written as follows,

$$DALN^k(x_i^k; \gamma^k, \beta^k) = \gamma^k \cdot \hat{x}_i^k + \beta^k, \tag{9}$$

where

$$\hat{x}_i^k = \frac{x_i^k - \mu_k}{\sqrt{\sigma_k^2 + \epsilon}}, \tag{10}$$

and

$$\mu_k = \frac{1}{d} \sum_{i=1}^{d} x_i^k, \quad \sigma_k^2 = \frac{1}{d} \sum_{i=1}^{d} \left(x_i^k - \mu_k\right)^2. \tag{11}$$

During training, DSLN employs a separate LN module for each dataset, ensuring dataset-specific normalization statistics and learned affine parameters. In the testing phase, for datasets used during pretraining, DSLN applies the corresponding dataset-specific LN, maintaining consistency with the training phase normalization. For novel datasets not included in pretraining, DSLN averages the outputs of all dataset-specific LN modules. This approach enables the model to generalize to unseen datasets by leveraging the collective learned normalization parameters.

## C  DETAILED RESULTS

### C.1  DETAILED RESULTS ON THE EFFECT OF VARYING NUMBER OF LAYERNORMS

Detailed analysis reveals that the performance of the `ProtoN-FM` model varies with the number of LayerNorms within each *ProtoNorm* layer across different datasets in both FD and HAR tasks. For the FD task, three LayerNorms yield optimal results, with an average accuracy of 70.33% and a Macro-F1 score of 67.80%, marginally outperforming other configurations. The IMS dataset shows improved performance with increased LayerNorms, peaking at 78.78% accuracy and 73.03% Macro-F1 score. Conversely, in the HAR task, aligning the number of LayerNorms with the number

of pretrained datasets ($\#D$) achieves the highest performance, with an average accuracy of 51.85% and a Macro-F1 score of 38.50%. This configuration's robustness is further evidenced by small gains observed in datasets like UCIHAR and USCHAD. While performance differences between configurations are relatively small, these findings underscore the importance of tuning the number of LayerNorms based on task complexity and dataset diversity.

Table 7: Performance comparison using different number of LayerNorms accross various datasets on both FD and HAR tasks. We calculate the Accuracy and F1-score (%) for each dataset. Best average performance results are **bolded**.

| Datasets | Accuracy | | | | Macro-F1 | | | |
|---|---|---|---|---|---|---|---|---|
| | 2 | 3 | 4 | $\#D$ | 2 | 3 | 4 | $\#D$ |
| IMS | 78.14 | 78.78 | 78.33 | 77.23 | 71.92 | 73.03 | 71.51 | 70.40 |
| UO | 66.83 | 68.56 | 67.60 | 68.48 | 66.10 | 67.93 | 66.77 | 67.72 |
| PU | 63.38 | 63.65 | 63.42 | 64.91 | 60.08 | 60.43 | 60.14 | 61.61 |
| Average | 69.45 | **70.33** | 69.78 | 70.21 | 66.03 | **67.80** | 66.81 | 66.58 |
| HHAR | 72.09 | 72.29 | 72.22 | 72.43 | 64.12 | 64.11 | 64.21 | 64.36 |
| SKODA | 26.01 | 24.26 | 23.48 | 25.56 | 16.27 | 15.93 | 15.50 | 16.94 |
| UCIHAR | 59.24 | 59.11 | 59.29 | 59.38 | 46.70 | 46.72 | 46.58 | 46.69 |
| USCHAD | 36.24 | 35.78 | 36.01 | 36.64 | 23.43 | 23.23 | 23.34 | 23.86 |
| WISDM | 63.48 | 60.85 | 62.69 | 61.25 | 43.70 | 42.56 | 42.87 | 42.67 |
| Average | 51.81 | 50.46 | 50.74 | **51.85** | 38.44 | 38.11 | 38.10 | **38.50** |

## C.2    DETAILED RESULTS ON THE GENERALIZATION ANALYSIS

Detailed analysis of generalization performance for both FD and HAR tasks reveals consistent improvements using the `ProtoN-FM` model over Vanilla pretraining across all datasets. In the FD task illustrated in Table 8, `ProtoN-FM` demonstrates significant gains in both accuracy and Macro-F1 scores. The IMS dataset shows an accuracy increase from 50.10% to 54.66%, with Macro-F1 rising from 42.69% to 45.12%. Similarly, the UO dataset improves from 68.23% to 72.88% in accuracy and from 67.64% to 72.40% in Macro-F1. On average, `ProtoN-FM` outperforms Vanilla pretraining in the FD task, boosting accuracy from 41.89% to 46.73% and Macro-F1 from 37.27% to 41.03%. The results in Table 9 show that for the HAR task, `ProtoN-FM` also exhibits superior performance. For example, in the USCHAD dataset, accuracy improves from 31.75% to 35.65%, and Macro-F1 increases from 19.49% to 23.30%. On average, `ProtoN-FM` increases accuracy from 47.78% to 49.12% and Macro-F1 from 36.32% to 37.49%. These results demonstrate `ProtoN-FM`'s enhanced generalization capabilities across diverse datasets compared to the Vanilla pretraining approach.

Table 8: Detailed results of generalization performance comparison of vanilla pretraining method and `ProtoN-FM` on FD Task. We calculate the Accuracy and F1-score (%) for each dataset. The best average performance results are **bolded**.

| Datasets | Accuracy | | Macro-F1 | |
|---|---|---|---|---|
| | Vanilla | `ProtoN-FM` | Vanilla | `ProtoN-FM` |
| IMS | 50.10 | **54.66** | 42.69 | **45.12** |
| UO | 68.23 | **72.88** | 67.64 | **72.40** |
| PU | 07.33 | **12.65** | 01.47 | **5.58** |
| Average | 41.89 | **46.73** | 37.27 | **41.03** |

Table 9: Detailed results of generalization performance comparison of vanilla pretraining method and `ProtoN-FM` on HAR Task. We calculate the Accuracy and F1-score (%) for each dataset. The best average performance results are **bolded**.

| Datasets | Accuracy | | Macro-F1 | |
|---|---|---|---|---|
| | Vanilla | ProtoN-FM | Vanilla | ProtoN-FM |
| HHAR | 70.18 | **70.51** | 62.18 | **62.64** |
| SKODA | 22.45 | **23.58** | 15.09 | **15.57** |
| UCHIAR | 57.64 | **58.49** | 45.38 | **46.29** |
| USCHAD | 31.75 | **35.65** | 19.49 | **23.30** |
| WISDM | 56.86 | **57.37** | 39.45 | **39.65** |
| Average | 47.78 | **49.12** | 36.32 | **37.49** |

