# OpenReview forum: "Towards Adaptive Time Series Foundation Models Against Distribution Shift"
_ICLR.cc/2025/Conference — ICLR 2025 Conference Withdrawn Submission_

### Official Review · Reviewer_3jEV · 2024-10-21

**Soundness:** 2
**Presentation:** 3
**Contribution:** 2
**Rating:** 3
**Confidence:** 3

**Summary:**

The paper introduces ProtoN-FM, a novel approach for addressing distribution shifts in time-series foundation models (FMs). The method integrates a prototype-guided dynamic normalization mechanism within the Transformer architecture, replacing traditional LayerNorm with ProtoNorm layers. These layers use learned prototypes to dynamically select the most appropriate LayerNorm for each input sample, ensuring better alignment with the diverse characteristics of time-series data.

**Strengths:**

The paper presents a novel approach to handling distribution shifts in time-series foundation models by introducing a prototype-guided dynamic normalization mechanism (ProtoNorm). This approach is original in that it leverages learned prototypes to dynamically select the most appropriate LayerNorm for each input sample, addressing a key challenge in time-series analysis that has not been sufficiently tackled by existing methods. Unlike traditional normalization techniques that apply static parameters, ProtoNorm adapts to intra- and inter-dataset variations, offering a fresh and effective way to improve model robustness and generalization.

The concept of dynamic normalization based on sample-to-prototype similarity is a creative extension of existing normalization strategies, and its integration within the Transformer architecture represents a meaningful innovation that broadens the scope of adaptive techniques in deep learning.

The paper is well-organized, with a clear problem statement and methodological exposition. The authors effectively communicate the motivation behind their approach, providing a logical progression from the identification of the problem (distribution shifts) to the proposed solution (ProtoNorm) and its integration within the Transformer framework.

The paper's demonstration of ProtoN-FM's robustness to distribution shifts, including scenarios with added noise, is particularly impactful. It suggests that the proposed method can be applied to practical, noisy, and diverse datasets, thus broadening its potential usage beyond controlled experimental settings.

**Weaknesses:**

1. While the paper demonstrates strong performance in classification tasks (FD and HAR), the proposed approach could benefit from expanding its applicability to other common time-series tasks such as forecasting or anomaly detection. Time-series data often involve tasks where the model needs to predict future values or identify rare events. It would be beneficial to show that ProtoN-FM can handle such scenarios effectively, which would highlight its versatility and broaden its potential use cases.
2. The method relies on maintaining multiple LayerNorms per ProtoNorm layer, which are selected dynamically using a prototype-guided gating network. While this approach is effective for a moderate number of datasets, it could lead to scalability issues when the number of datasets becomes very large, as the gating mechanism would need to handle a much larger set of prototypes.
3. The method’s reliance on learned prototypes that act as centroids for different data distributions raises the risk of overfitting to the pretraining datasets. This could lead to suboptimal performance when the model encounters out-of-distribution data during fine-tuning or deployment. Although the model demonstrated robustness in the experiments, it would be valuable to understand how it behaves under more significant distribution shifts.
4. The current paper does not provide a deep exploration of how the prototypes evolve during training and how they adapt to various data distributions. Understanding this behavior is crucial because it would help determine whether the prototypes are capturing meaningful patterns or simply acting as a form of overfitting.
5. The paper could benefit from a more detailed justification of certain design choices, such as the use of Euclidean distance for prototype selection or the specific structure of the prototype update mechanism. While the current setup works, there may be scenarios where other distance metrics or update strategies could be more effective.
6. While ProtoN-FM has been shown to improve performance across different datasets within the same task domain (FD or HAR), there is less emphasis on how well it generalizes across completely different task domains. For example, if the model is pretrained on a mix of FD and HAR datasets, can it generalize effectively to a new domain like weather data or financial time series?

**Questions:**

see Weaknesses

---

### Official Review · Reviewer_JrdL · 2024-11-03

**Soundness:** 3
**Presentation:** 3
**Contribution:** 3
**Rating:** 5
**Confidence:** 3

**Summary:**

The paper introduces ProtoN-FM, a novel approach to handle distribution shifts in time series foundation models through prototype-guided dynamic normalization. The method replaces standard LayerNorm with a prototype-based mechanism that dynamically selects normalization parameters based on learned prototypes representing different data distributions. The approach is validated on fault diagnosis and human activity recognition tasks.

**Strengths:**

The paper presents a novel adaptive normalization strategy aimed at addressing the distribution shift problem in time series analysis. By employing a prototype-guided dynamic normalization mechanism, it significantly enhances the model's generalization ability across different datasets. This innovative approach is the first of its kind in the field of time series, showcasing the authors' unique research perspective. Experimental results demonstrate that the proposed ProtoN-FM model outperforms existing methods on multiple benchmark datasets. Detailed ablation experiments ensure the reliability of the research findings. The paper is well-structured and logically rigorous, with figures effectively supporting the arguments, making complex results more understandable. As time series data is widely applied, this research outcome provides new insights into dealing with data distribution changes, holding significant academic value and practical significance.

**Weaknesses:**

- The paper conducted experiments on multiple datasets, but the diversity and representativeness of the selected datasets may be insufficient. It is recommended to expand the scope of future experiments to cover more types of time series datasets, such as finance, healthcare, or climate data.
- Introducing a prototype-guided dynamic normalization mechanism may increase the complexity and computational cost of the model. The authors should consider optimizing computational efficiency and providing a more detailed cost analysis. Report training times, memory usage, or parameter counts compared to baseline models, and analyze the computational complexity of their method relative to standard LayerNorm.
- Although various baseline methods were compared, not all the latest technologies were covered. It is suggested to add comparative experiments to fully assess the relative advantages of ProtoN-FM.
- At the same time, there is a lack of in-depth theoretical analysis on the reasons for the model's performance improvement. It is recommended that the authors explore the specific impact of prototype-guided dynamic normalization on the learning process and generalization ability. How do the learned prototypes relate to dataset characteristics? For ablation studies, isolate the effect of different components of the dynamic normalization process.
- The interpretability and user-friendliness of the model should also be taken into consideration, and it is suggested to discuss visualizing the prototype and normalization process. Visualize how samples map to different prototypes. Analyze how the normalization parameters change for different inputs to help users understand the model's decision-making process.
- Limited theoretical foundation for prototype effectiveness
- Missing comparisons with recent adaptive normalization methods
- Computational overhead not analyzed
- Limited to classification tasks
- Choice of prototype number needs better justification
- Scalability analysis missing
- Some implementation details unclear
- Limited analysis of failure cases

**Questions:**

1. The paper has some issues with the rationality of dataset selection. The time series datasets used may not adequately represent characteristics from different fields. It is recommended that the authors include more diverse datasets from areas such as finance, healthcare, and climate in future research to verify the model's universality and robustness.
2. At the same time, introducing a prototype-guided dynamic normalization mechanism may increase the model's complexity and computational cost. It is suggested to provide a detailed analysis of computational efficiency and optimize the model structure to ensure its feasibility.
3. Regarding the depth of theoretical analysis, it is recommended to explore the specific impact of dynamic normalization on the model's learning process to enhance the depth and persuasiveness of the research.
4. In terms of interpretability and user-friendliness, it is suggested to use visualization tools to help users understand the model's decision-making process.
5. How does the computational complexity compare to standard LayerNorm?
6. Can you provide theoretical justification for the prototype learning process?
7. Why not evaluate on forecasting tasks?
8. Can you provide comparison with recent adaptive normalization approaches?

---

### Official Review · Reviewer_qUza · 2024-11-04

**Soundness:** 1
**Presentation:** 2
**Contribution:** 2
**Rating:** 3
**Confidence:** 3

**Summary:**

The paper addresses the challenge of applying foundation models to time series data by introducing a domain-aware adaptive normalization strategy. This approach uses prototype-guided dynamic normalization to better align pretrained representations with downstream tasks, improving fine-tuning performance and mitigating distribution shift issues. Extensive experiments show its effectiveness on real-world time series datasets.

**Strengths:**

1. The proposed ProtoNorm method is transferable and can be applied to various models using LayerNorm, not limited to a specific architecture.

2. The approach of dynamically selecting the most suitable LayerNorm to adapt to different input distributions is simple and effective, without significantly increasing parameters or inference costs.

3. Experiments show stable performance improvements over direct pretraining and finetuning when using PatchTST as the model backbone for downstream time series classification tasks.

**Weaknesses:**

1. Backbone choice: The paper uses PatchTST as the model backbone for both pretraining and downstream time series classification tasks. This choice may be insufficient since, while PatchTST is SOTA in time series forecasting, it seems that it is not among the top-performing models for time series classification and representation learning. In practice, using a relatively weak baseline could make even minor perturbation strategies appear more effective. To highlight the significance of the proposed method, I recommend applying ProtoNorm to time series classification models with top performance and demonstrating the potential improvements. Alternatively, if the authors find that ProtoNorm cannot be applied to these existing SOTA methods, I suggest comparing the performance of these SOTA models directly with PatchTST before and after applying ProtoNorm. Even if ProtoNorm-enhanced PatchTST does not fully surpass the SOTA models but comes close, it would better highlight the significance of this work.

2. Comparison with related methods: The paper introduces a pretraining method to enhance time series representation and improve performance on datasets with significant distribution differences. The authors should compare ProtoNorm with related methods to demonstrate its effectiveness. This could include comparisons with simple time series data augmentation techniques, other contrastive learning-based pretraining methods, or other adaptive normalization methods like RevIN. For instance, comparing with RevIN or combining ProtoNorm with RevIN to observe performance gains would be valuable. This comparison is crucial because methods like RevIN add minimal computational cost, whereas the proposed ProtoNorm requires pretraining stage that significantly increases training costs. Such comparisons would help users decide whether the additional cost of ProtoNorm is justified for their needs.

3. Details on the gating network: The paper mentions a gating network that projects the input into the feature space for the computation of the distances with existing prototypes. However, the structure of this gating network is not mentioned in the paper. Is it a gated MLP? Does the output of the gating network feed into subsequent model layers as features? If not, it is unclear how this network receives gradients for parameter training. Clarifying these aspects would strengthen the method’s explanation.

**Questions:**

1. How are the LayerNorms selected and when does this selection occur? Is the distance between the input and prototypes calculated once for the entire model backbone with a single decision made for all LayerNorms? Considering PatchTST’s structure, where a time series is divided into multiple patch tokens fed into encoder-only Transformer layers, does ProtoNorm select different LayerNorms for each patch token at each layer? The latter approach seems to offer greater expressiveness, but the current description in the paper is unclear for which one is used, making it difficult to understand the method details.

2. How is the number of prototypes determined? It seems like this should relate to the diversity of the pretraining dataset—should more diverse datasets correspond to more prototypes? Scaling up the number of prototypes and corresponding number of LayerNorms should theoretically improve expressiveness harmlessly, as the model can ignore irrelevant LayerNorms if the input distribution does not fit them. However, Figure 4 shows that increasing the number of LayerNorms leads to performance drops. Why does this happen?

3. Using an orthogonal loss to optimize feature space separability with a model dimension of $ d = 128 $ and $ n = 3 $ LayerNorms seems somewhat ineffective, as the likelihood of three vectors being non-orthogonal in a 128-dimensional space is very low. Yet, experiments show this constraint significantly improves performance. Is this because, even with orthogonal initialization, the vectors still tend to collapse without this constraint, making it necessary despite having only three prototype vectors?

I may raise my score if my concerns get addressed.

---

### Official Review · Reviewer_8yEP · 2024-11-04

**Soundness:** 2
**Presentation:** 2
**Contribution:** 2
**Rating:** 5
**Confidence:** 3

**Summary:**

The paper introduces ProtoN-FM to improve the performance of foundation models (FMs) on time series data with diverse distributions. Traditional LayerNorm is replaced by a dynamic normalization technique within the Transformer architecture that learns prototypes representing different data distributions, allowing per-sample normalization based on sample-to-prototype similarity. ProtoN-FM addresses significant challenges in handling distributional shifts during pretraining and fine-tuning, enhancing alignment between pretraining representations and downstream tasks. Experimental results demonstrate ProtoN-FM’s effectiveness across tasks such as fault diagnosis and human activity recognition, surpassing standard baseline approaches.

**Strengths:**

-The use of prototype-guided normalization to dynamically adjust to different data distributions presents a fresh approach for improving FM robustness on TS data.

-The model is tested across several real-world TS datasets, showing superior fine-tuning performance and robustness to distribution shifts.

- ProtoN-FM provides insights into the impact of distribution shifts, an important consideration for TS applications in high-stakes domains like healthcare and finance.

**Weaknesses:**

- While the paper includes experiments on prototype update parameters, additional exploration of the impact of different initialization methods for prototypes would enhance understanding.

- Although tested on FD and HAR tasks, the method could benefit from experiments on broader time series forecasting or anomaly detection tasks to strengthen the generalizability claims.

**Questions:**

The paper demonstrates results on classification tasks. Could the authors comment on the potential to extend ProtoN-FM to forecasting and anomaly detection, where time dependencies are particularly critical?

How critical is the self-supervised pretraining phase with contrastive learning to the success of ProtoN-FM?

---

### Note · Authors · 2024-11-22

I have read and agree with the venue's withdrawal policy on behalf of myself and my co-authors.